# Effectiveness of Mobility and Urban Sustainability Measures in Improving Citizen Health: A Scoping Review

**DOI:** 10.3390/ijerph20032649

**Published:** 2023-02-01

**Authors:** Carmen Fernández-Aguilar, Marta Brosed-Lázaro, Demetrio Carmona-Derqui

**Affiliations:** 1Faculty of Economic Sciences, International University of Isabel I of Castilla, 09003 Burgos, Spain; 2Department of Applied Economics, University of Granada, 18071 Granada, Spain

**Keywords:** sustainable mobility, mobility plans, public health

## Abstract

Background: The relationship between mobility and health has multiple dimensions, and the mobility model can be considered a public health intervention. Increasingly, mobility in cities is oriented towards incorporating sustainability criteria; however, there are many very diverse measures that cities carry out in terms of mobility and urban sustainability, and in many cases, these do not receive subsequent evaluation and/or study to analyse their effectiveness or impact. Currently, the literature does not offer any updated review of the measures applied in the different communities and countries. Aim: To carry out a panoramic review of the measures implemented in the last 5 years to analyse which ones report a greater effectiveness and efficiency in health. Results: After applying the exclusion criteria of the study, a total of 16 articles were obtained for evaluation. The measures applied in terms of sustainability are grouped into four subgroups and their subsequent evaluation and possible impact on public health is analysed. Conclusions: The present study found a large heterogeneous variety of sustainability measures in local settings around the world, which seem to reflect positive impacts on population health. However, subsequent evaluation of these measures is inconclusive in most cases. Further research and sharing across macro-communities are needed to establish universal criteria.

## 1. Introduction

In recent decades, profound social, economic, and technological changes have led to a new model of urban mobility. This model, which is tending to be implemented globally, is characterised by an increase in the average distances travelled, changes in the reasons for travel, and changes in the location of productive activities [1].

The distance separating the places where different economic and social activities are carried out has continued to grow in recent decades because of technological and organisational advances. The increase in the speed–distance binomial has allowed the “technological distance” between two points to replace geographical distance, and a large part of the time gained by reducing the working day has been devoted to commuting [2].

On a global scale, we are suffering from atmospheric pollution and global warming due to vehicle emissions, and destruction of natural areas due to the continuous expansion of roads. On a local scale, the configuration of metropolitan areas is shaping an intensive model in the use of private vehicles, cities consume 70% of the planet’s natural resources and, in addition, it is expected that within 30 years, more than half of the world’s population will be living in urban environments. For these reasons, it is necessary to adapt cities to a sustainable model that allows the coexistence of their inhabitants and their development without aggravating the social and climatic problems in which mankind is immersed [3,4].

As a result, healthy lifestyles, as well as public policies and measures whose social interest is the generation of sustainable urban spaces and mobility have emerged as an important issue in contemporary society, being a fundamental aspect in the public health of the population. In view of this circumstance, governments and institutions promote the revitalization of public spaces, which encourages participation in physical activity and the development of human interrelationships [5]. Thus, an active lifestyle, including walking and cycling, has become a key factor in reducing the impact of chronic diseases, obesity, and/or coronary problems linked to human behaviour, such as those caused by lack of physical exercise [6]. Similarly, when these activities are used as a means of transportation, they promote a less congested urban environment and reduce greenhouse gas emissions, which cause climate change. This has a positive impact on human health [7].

The field of study related to urban mobility and active transportation in cities has emerged as a relevant area of research. The growing interest in caring for the environment has produced a sociological change in broad sectors of society, which is summarised in the implementation of different measures and policies, including polycentric 15-min cities, improved accessibility to public transport, creation of corridors or green lungs, pedestrianisation of streets, bicycle lanes, etc. [8]. Recently, these measures have been distinguished as push or pull measures and policies. This description differentiates between those measures that promote or encourage a sustainable use of the environment, or the improvement of a habit. In addition, these measures are aimed at limiting and/or prohibiting the use of a specific type of resource [9].

However, and despite the wide range of measures carried out in relation to urban and sustainable mobility, empirical evidence on the evaluation and/or measurement of these measures seems to show great deficiencies, due to the lack of guidelines in the measures themselves that allow the necessary evaluation of whether or not they are effective, whether citizens perceive them as useful, or whether they have positive repercussions on the health of the population [7].

Therefore, the aim of this paper is to highlight the information and measurements available on this type of urban and sustainable mobility plan implemented in recent years, with the ultimate goal of providing a panoramic spectrum on the most effective models of urban and sustainable sustainability in terms of transferring public health to the population. By accomplishing this task, the aforementioned gap found in the literature, in terms of guidelines, could be fulfilled, providing useful information to the policymakers, thus allowing a first step in the development of possible universal criteria and/or protocols for action, as support tools for local levels, in the implementation of such measures. 

Other reviews have made valuable contributions to this debate in recent years in an attempt to unify a useful knowledge base in this regard; however, the heterogeneity of the measures analysed and of the measurements, as well as the local characteristics of most interventions, are a recurrent obstacle to this task. On the other hand, these studies tend to focus on a very specific type of measure, with the result that there are few studies in the academic literature that allow comparison of different types of measures. For example, Stappers et al. [10] were only interested in studies that analysed the effect of changes in city infrastructure. Nieuwenhuijsen et al. [11] only collected studies that analysed the impact of green spaces on the health of citizens in urban contexts. Another more recent review by Nieuwenhuijsen [12] addressed studies on CO_2_ emissions, while the meta-analysis by Atkinson et al. [13] focused on NO_2_ emissions. 

The rest of the paper is organised as follows: the next section addresses a deep explanation of the methodology. Section 3 describes the main results found. Section 4 is a discussion of the results, and finally, Section 5 consists of a summary of the main conclusions, which works as a guideline for policymakers.

## 2. Materials and Methods

The scoping review was carried out following the methodology proposed by Arksey and O’Malley [14], which is used as a reference in the field. Overview reviews are part of a distinct group of review models that are characterised by answering broader questions and allow analysis of the evidence through a balance of breadth and depth of topic.

This system is made of six steps of working: 

### 2.1. Identify the Research Question

The particular question selected for this overview review is as follows: What has been the public health impact of sustainable urban mobility measures in recent years?

### 2.2. Search Strategy: Identify Relevant Studies

The research was performed on databases in the Web of Science during the period 2018 to 2022. Articles published in English, Spanish, French, and Portuguese were included. This search was carried out by two experts in scientific literature searching (DC, CF). The search terms used, and their respective results are listed in Appendix A, according to the taxonomy of each database. This search strategy aims to identify papers published in recent years that address the impact of sustainable urban mobility measures on public health, understood in a broad sense, which includes both direct and indirect measurements of public health impact.

We initially reviewed all references obtained, both empirical studies and non-empirical studies. Python language was used to process the files with the bibliographic records. Duplicates were removed to obtain a preliminary selection of studies.

The screening process was carried out in three stages: First, the titles and abstracts of all articles were reviewed by two authors independently (DC and CF) to determine eligibility. In the second phase, both extensively reviewed each provisionally selected article using an ad hoc review protocol. The final list of articles was sent to an expert (MB), to solicit the opinion on the deletion of articles that are not considered relevant and for the inclusion of new articles. Review by a third reviewer is one of the main keys to the overview review, helping to verify the selection of studies.

The final bibliography was reviewed for thematic analysis and synthesis by all the authors of this manuscript.

### 2.3. Selection of Studies

This inclusion and exclusion criteria were applied. 

This study is focused on papers that fulfil both of the following: to be an empirical analysis and to provide sustainability policies based on previous empirical results. Therefore, systematic reviews are excluded from the focus, as well as papers based on COVID mobility measures and policies that have an impact over limited stakeholders. Likewise, studies supported by experts’ opinions are not considered in the final selection among others. Although reviews on the subject were excluded, they were used in the theoretical framework of the study and are thus referenced.

These criteria determine the choice of those studies that can provide universal and applicable measures in different settings and time periods, on which the analysis of effectiveness and efficiency in terms of public health improvement will be carried out.

### 2.4. Data Extraction and Categorisation

A standardised formula to collect and extract information from each article was designed. This format was piloted, refined through an internal peer review process, and finally applied to all the selected studies by two of the authors (DC and CF). Of these studies, 35% were selected for repeat extraction of information by the other author (MB). This method was used to check the accuracy of the selection of information in the chosen extraction formula. Minor discrepancies were resolved by consensus between the four reviewers. The complete collection process is depicted in Figure 1. 

### 2.5. Quality and Replicability of Studies

To facilitate understanding of the selected studies, given the heterogeneity of their conditions, study population, methodology, and setting, each study was reviewed using the simplified TIDieR criteria (Appendix B). 

The resulting 12-item TIDieR checklist (short name, why, what (materials), what (procedure), who provided, how, where, when and how much, adaptation, modifications, how well (planned), how well (actual)) improves the reporting of interventions and makes it easier for authors to structure descriptions of their interventions, for reviewers and editors to evaluate descriptions, and for readers to use the information [15].

### 2.6. Summary and Reporting Information

The information obtained from each selected article was synthesised as described in Table 1. The variables of analysis selected for the evaluation of the articles were, in order of appearance in the table: authors, year of publication, location of the study, the method of analysis applied, the population or study subjects, the number of participants or sample size, the sustainability and mobility measure analysed, the post-evaluation of the effectiveness of the measure, the analysis subject to the health impact of the measure analysed, and the transferability of the study, and/or the case to other settings or places (Appendix C).

## 3. Results

### 3.1. Overall Descriptive Results

A total of 546 articles were initially obtained. After the review process and application of the criteria, 16 articles were finally selected. All were published between 2017–2022, with different locations across the globe. It should also be noted that all selected studies were developed or funded by research centres and/or universities, without finding any study subject to any specific governmental institution. The populations and fields of study present a wide heterogeneity among them, showing quantitative and qualitative studies. 

In the analysis applied to extract information on the variables subject to study, the results are summarised as follows. The numerical data can be seen in a synthesised form in Table 1.

The measures carried out are grouped into four different groups, according to the type of measures identified in the selected studies, as follows: measures dedicated to the promotion of cycling (A) (37.5%), interventions dedicated to the improvement of the structure and organisation of public spaces and streets (B) (18.75%), to reductions and/or optimization of the use of motorised vehicles (C) (31.25%), and in plans for mixed measures of urban sustainability (D) (12.5%). 

Regarding the results or impact on health, 13 of the 16 articles (81.25%) seemed to indicate that the application and measures can generate a health impact for the population. Three of the articles did not mention the possible health implications of the measures developed. 

Regarding transferability, i.e., the possibility of transferring this type of measures to other places or larger spaces, only seven of the articles (43.75%) reflect this possibility. 

Regarding the post-implementation evaluation of the measures, which is the subject of this review, half of the articles (50%) made a post-implementation evaluation related to the effectiveness and satisfaction with the use and development of the measure. 

Within the definition of push and pull measures (those that seek to incentivize and/or motivate to perform an action, as opposed to prohibiting or limiting an action), we found that the bulk of the measures (75%) were of the push type, seeking to incentivize and/or motivate users to perform specific habits and actions.

### 3.2. Specific Synthesised Results

The measures that have been grouped in this study as type A, B, C, and D measures, are developed and specified below in a panoramic view, together with the benefits and impacts indicated therein; measures A and C report the highest percentage of use.

Jung et al. [16] analysed the impact of Seoul’s Design Street Project on pedestrian satisfaction and traffic on the refurbished streets. This project consisted of a series of improvements to both pedestrian walkways and public spaces, signage, and other relevant street elements, which were implemented between 2007 and 2010. Their results suggest that the improvements to the physical elements of the streets were effective in improving the satisfaction of citizens passing through them but had no effect on the number of people walking on them.

In turn, Marcheschi et al. [17] reported on four car-free street experiments in Malmö and Gothenburg, two Swedish cities, and elaborated an analysis of how citizens perceived the changes that have taken place on the streets. The design of the experiments focused on measures that encourage pedestrian traffic instead of cyclist traffic and took place between 2016 and 2019. They highlight the importance of psychosocial processes when designing and implementing motor vehicle traffic restriction measures, as that they may be a determinant for the acceptability of the measures.

Zhang et al. [18] developed a model to optimise the layout of stations employing bike sharing services. To illustrate their functionality, they employed data collected in Satagaya Ward, Tokyo, via GPS in 2012. The data comprised trajectories of individuals walking, cycling, or driving. The authors used three scenarios to estimate the impact that optimising stations would have on emissions, each with a different ratio of car trips replaced by bicycle trips: 100%, 50%, and 10%. They concluded that their optimization method can potentially reduce emissions by more than 3 thousand tons and improved on forecasts made by other optimization models proposed previously.

Maisel et al. [19] evaluated the perceptions of pedestrians on the implementation of the Complete Street project in the city of Williamsville, New York, between 2018 and 2019. Complete Street projects aimed to improve streets to promote active transportation and public health, among other things. They conducted surveys before (2015) and after (2019) the selected street modifications were made. They found evidence that the changes improved citizen satisfaction but did not necessarily increase physical activity. In fact, they highlight that individuals perceived traffic to run faster, which could be counterproductive.

Egiguren et al. [20] performed a quantitative health impact assessment, with data from 17 countries from the Americas, Europe, Asia, and Africa, to estimate the potential impact on emissions on achieving the targets set by the projections of a previous study (Mason et al., 2015) for 2050. Models were developed for two scenarios: in the first, assuming optimal conditions and 8% of bicycle trips replacing car trips, a reduction of 18,589 premature deaths was estimated; in the second, assuming the same conditions and 100% replacements, the estimate amounts to 205,424 premature deaths in the set of countries analysed.

On the other hand, Otero et al. [21] examined the main bike-sharing systems (with more than 2000 bicycles) located in 12 European cities and evaluated their impact on citizen health using a quantitative health impact assessment. They estimated that these systems avoid, on average, 5 deaths per year, and 73 deaths among all the systems analysed, saving €18 million. Although the use of bicycles also carries some health risks, such as an increase in cyclist fatalities, the results suggest that the potential benefits outweigh the potential harms.

Tao et al. [22] analysed the Dockless Bike-Sharing Service, which is spread across China; this included several improvements to make the use of bikes easier and more attractive, such as the GPS tracking systems and mobile payment among others. The case was studied in Shanghai, the largest DBS market in the world, and focused on comparing the impact of different transport modes. The potential benefits of DBS from emission reduction were assessed as 0.023 and 0.040 CNY/passenger-km, 5.4 min in terms of time savings in journeys of 3 km, and finally, the economic benefit was calculated as 0.085 CNY/passenger-km in high utilisation. The study also evaluated the negative impact of DBS systems in terms of public space occupation; however, this was minor considering the positive effects. 

Chatziioannou et al. [23] aimed to assist in the implementation of a comprehensive sustainable mobility and transport plan in Mexico City to improve the quality of life of citizens. To this end, they constructed a global index in which they analysed the sensitivities of each indicator using MICMAC software, which allowed them to organise them according to the quantitative impact they have on urban mobility. The result of their research indicates that the most relevant variable in terms of sustainable mobility was proximity to points of interest followed by block size and shape and public transport coverage. The strength of quantifying the elements allows this study to set a guideline for the implementation of sustainable mobility plans applicable in other environments. 

Mueller et al. [24] estimated the health impact of the Superblock Model, implemented in Barcelona, Spain, using a quantitative health impact assessment. This model was composed of numerous measures aimed at reducing motor vehicle traffic in strategic areas of the city. The analysis considered changes in the population’s physical activity, air pollution, traffic noise, green spaces, and the urban heat island. A total of 667 premature deaths could be avoided annually after implementing the 503 planned superblocks. Reducing emissions was the main factor affecting the reduction of premature deaths, followed by changes in noise, heat, and green spaces.

Mateu and Sanz [25] examined a comprehensive set of cycling promotion measures implemented in the city of Valencia, Spain, from 2016 to 2020. They used data collected through sensors connected to strategic points of bicycle lanes throughout the city. Their results showed that the city lacks some critical elements, such as bicycle parking areas, integration of bicycles with other transports, and end-of-trip facilities. Despite this, there was an increase in traffic intensity when controlling for seasonality. The authors highlight the success of the city’s bike-sharing programme.

Cerutti et al. [26] carried out a study of how the bike-sharing system is perceived in a medium-sized smart Brazilian city, Passo Fundo, and detected the motivations that lead residents to use this means of transport. To do this, they conducted a survey among residents, users and non-users of the system, and the responses were analysed at a descriptive level; more sophisticated techniques, such as the use of Cronbach’s alpha factor, varimax rotation for factor analysis, and ANOVA, were used to detect differences between the group of users and non-users. This allowed them to obtain a list of the 10 most relevant variables in the decision to use the bike-sharing system, with the variables related to “Health and environment” having the greatest impact, and to use the interconnections found to guide the development of smart cities and the implementation of sustainable policies. 

The work of Billions et al. [27] is the first part of an unfinished research project focused on analysing the key elements of a smart region, specifically those related to mobility. The environment studied was the Western Visayas region in the Philippines, specifically the implementation of “Sustainable Technology-Assisted Route Planning for Region”, taking socio-economic data, coverage maps, and inventories of public transport and communication infrastructures built or in progress. As a result, two nuclei of smart cities were detected in the region and the variables with the greatest impact on the future design of the transport system implemented in the region, which will be used in the second part of the research where the effect of smart cities on the development of the region will be evaluated.

In Reche et al. [28], the objective was to study the negative aspects that excessive vehicle use generates at an environmental and health level in the population, as well as to quantify the benefit of carrying out certain mobility interventions in such a way that these results are considered in decision-making and the design of healthier cities. By taking data on pollution levels in 12 European cities and using a long-linear model, the impact on health in the form of increased mortality was obtained. After the application of different measures, a decrease in premature mortality of 1.7% was estimated due to the decrease in exposure to PM2.5 and NO_2_ as the main result, which encourages the study of the implementation of more ambitious measures in terms of sustainable mobility.

Cambra et al. [29] performed an ex-ante and ex-post analysis of the Eixo Central project, a programme aimed to improve the walkability conditions in Lisbon. Pedestrian flow data was collected by using the gate method during working days and a survey was conducted in the Eixo Central area, before and after the implementation of the changes, which made the paired t-test a technique suitable with the nature of the study. The main conclusion of this study is the scale of the environmental intervention; larger interventions had a deep impact on population manners regarding walkability, which is not the same for micro-interventions. 

The holistic program Madrid City air-quality plan 2020 was analysed by Izquierdo et al. [30], which provided very interesting results regarding the impact of this type of measure on health. By taking data of annual mean concentrations (μg/m^3^) of PM2.5, NO_2_, and O_3_, as well as data on daily mortality from the Regional Statistical Office, the research applied Standardised HIA methods (health impact assessment). The results were very determinant; through a reduction in the annual mean PM2.5 concentration of 0.6 μg/m^3^ and 4.0 μg/m^3^ for NO_2_, the annual number of deaths caused by long-term exposure (95% CI) that could be postponed by the expected air-pollutant concentration reduction was on average 88 for PM2.5 and 519 for NO_2_, having the largest impact in the centre of the city.

Barcelona’s Superblock Model, focused on reducing vehicles to improve air quality, was analysed by Rodriguez et al. [31]. The methodology used is a multi-scale modelling chain (VML-HERMESv3) and the mesoscale air quality system allowed the evaluation of each measure individually and collectively leading to an interesting result; only when several measures were combined, the effects over emissions were significant. Applying LEZ, TUP, and SPB NOx emission reductions reached 30% and NO_2_ emissions 25%. Therefore, the authors recommended the use of this kind of analysis to forecast the impact of each traffic management strategy before the implementation. 

## 4. Discussion

Environmental sustainability has become an essential part of urban performance reporting and planning. Population pressure and proximity to resource limitations, current or anticipated exogenous environmental impacts, and a greater sense of global responsibility have catalysed those planning or governing cities to be guided by studies of environmental sustainability.

This scoping review aims not only to identify and synthesise locally implemented measures for sustainable urban mobility, but also to analyse and study their evaluation and effectiveness in terms of public health. In this sense, the review sheds light on the different measures that are being implemented in different cities around the world, but at the same time, it shows the heterogeneity of these measures, presenting a wide range of urban and sustainable measures. This heterogeneity is even more remarkable in terms of the local areas that analyse the effectiveness of the measure after implementation. In this review, only half of the studies analyse the impacts and results of having implemented the specific measure. Most of the studies (81.25%) seem to indicate that such measures have a positive impact on the health of the population, but only 50% of them are subject to a post evaluation of this impact. This result is in line with the results of previous reviews on similar topics [11].

The information gap persists in the difficulty of performing a health impact analysis, mainly due to the inherent biases of such a study, but also in the difficulty of applying homogeneous measures between local settings, given that these measures are inherently subject to the specific characteristics of both the area of application and the culture and qualities of the population [32].

This scoping review groups the different measures applied in each of the studies into four subgroups: A = measures to promote the use of bicycles; B = interventions to improve streets and public spaces; C = reduction and/or optimization of motorised transport; and D = plans for mixed urban measures, which in turn are grouped into push or pull type measures.

A considerable number of studies regarding the bike sharing system implemented in cities have been found during this research, as possible measures of improvement. Undoubtedly, the interest in urban cycling is increasing, especially over the last 10 years, due to the health advantages it entails. The benefits of bike sharing are flexible mobility, emissions reductions, physical activity benefits, congestion reduction, and fuel use (Shaheen et al., 2010). In addition, it does not produce noise. These positive aspects drive innovation in systems like the dockless bike-sharing system, the core of many studies specifically in China, which has the aim of boosting the use of bicycles by making it easier for citizens. 

The second series of measures found in literature are related to interventions to improve streets and public spaces, linked to the concept of walkability. These actions are supposed to be ways of producing environments with lower density, promoting physical activity, extent of street connectivity, increasing the green spaces, and ensuring equality and inclusion among all people. The costs related to the design of the cities are high and they do not seem to have an impact on the traffic on the roads where the improvements are implemented [16,19]; however, they may be justified in terms of improvements in walking conditions. Available evidence suggests that the effect of such interventions is often dependant on the context in which they are delivered and received [10], but the two studies included in this review make no particular reference to this factor.

The main group of pull measures are focused on reducing the use of motorised vehicles since the positive consequences are direct and undeniable; by reducing traffic lanes dedicated to private vehicles and boosting pedestrian and public transport, a decrease in greenhouse gas emissions is achieved. Simultaneously, the health benefits of reducing air pollution have been assessed in several studies; therefore, a relationship between car-free and health impact can be asserted. Somehow, this kind of measure is linked to the former since the outcome of both can be assessed in terms of the improvement in sustainable mobility, and the impact over health is similar [33]. 

Finally, due to the heterogeneity of cities and metropolitan areas and the huge number of challenges in terms of urban development that should be faced, it is common to find assessments of Integral Plans, which combine different types of measures, with the aim of having a holistic view of the task, satisfying stakeholders from different sectors, including environment, energy, land-use planning, or healthcare. This kind of plan used to involve push and pull measures. 

Technological advances in sustainable transportation, energy, and waste management; limiting the use of motorised transport; designing buildings and cities that reduce or improve transportation; mobile applications; creating green spaces; bike-sharing systems; and big data analytics offer opportunities to improve the health of urban populations while reducing their pollution and carbon footprint [34].

However, technological solutions alone are unlikely to solve the persistent environmental, social, and health problems facing 21st century cities around the world. To achieve long-term, sustainable solutions, a concerted and common action is needed by key stakeholders—the government, planners, health professionals, researchers, businesses—as well as individual citizens. To this end, it is necessary to carry out an in-depth analysis of the measures carried out, as well as of the holistic approaches to such measures, to provide, in a comprehensive manner, protocols and support resources that facilitate the implementation at local levels of those measures that are most suited to their characteristics and needs. 

The challenges of translating and unifying all these applications into a homogeneous system for generating a sustainable urban environment that effectively improves the public health of the population are complex, requiring a place-based understanding of the potential impacts and opportunities for health improvement that considers the social and cultural context of cities around the world. Most importantly, this requires effective public, political, and business engagement to ensure that scientific findings are translated into viable long-term solutions. Research such as that conducted in the present review can help shed light on the different options and analysis of current measures for sustainable urban mobility.

## 5. Conclusions 

The objective of this study is to make a scoping review of the literature production of the last 5 years on mobility and urban sustainability policies, in order to establish a guide of good manners and to detect the limitations of the analysis already done. The methodology selected allows us to answer wide questions regarding the specific area of analysis, keeping the rigour of systematic reviews but being more flexible. This kind of method requires the researchers to carry out a critical analysis of the available information and it is suitable with the core of the study since there is a wide variety of sustainable urban mobility measures and applications in different local areas. However, most assessments focus on case studies, which makes it difficult to draw conclusions about the effectiveness of the measures.

It has been found that of the studies included in the review, 75% analysed push-type measures, whose objective is to encourage and motivate the population to participate, suggesting that this type of measure is implemented more frequently. On the other hand, about 82% of the measures reflect a positive impact on health, but only 50% of them are subject to a post evaluation of this impact. In addition to this, in a vast majority of cases, several measures from different natures are applied simultaneously. Commonly, it is asserted that the effectiveness of the measures is greater this way. However, there is a lack of methods to evaluate the isolated impact of each initiative to hierarchize and prioritise in case of necessity. 

All this considered, further evaluation and analysis of the application, effectiveness, and impact of the measures is needed in order to better adapt and use them in the areas in which they are applied.

This review has some limitations. Firstly, the vast majority of empirical nature studies are specific cases of specific cities. It is common to find the analysis of the same measure in two cities with a difference of several million inhabitants, as well as different climatic conditions, orography, and culture. Obviously, the conclusions obtained in a city should not be applied in another city entirely; however, the challenge faced is to apply them in cities with similar features. To give a general extent and applicability to this kind of studies, as argued in previous works [12], it is recommended to add a comparative analysis where the same methodology was replicated with data from another city of similar characteristics, matching this way the conclusions obtained. Secondly, this review has a smaller number of articles than other reviews conducted in previous periods. This may be because the period chosen for the selection of articles covers the beginning of the SARS-CoV-2 health crisis and its development up to the present day. Articles on the pandemic and possible ways to mitigate its effects through urban planning have dominated much of the literature on sustainable mobility and public health since 2020. These papers are outside the scope of this review. The mobility restrictions put in place by authorities to contain the spread of the virus are a disruptive element that introduces new logics in urban planning and requires its own conceptual framework. While this field of study is interesting, it may have interfered with the development of other work on mobility and public health that did not include an analysis of the pandemic, resulting in a reduced flow of publications on this topic in recent years.

The need to unify criteria and establish protocols for measuring the impact of the measures analysed is evident both in this review and in previous similar work. Future reviews and studies in this area should prioritise the objective of meeting this need to achieve a common framework of analysis that provides a robust knowledge base for public decision-makers.

Despite these limitations, this study opens the path of the succeeding research work, which should take the direction of filling up the aforementioned literature gap by developing an empirical analysis of the specific measures and a ceteris paribus analysis of those measures applied simultaneously, as well as by defining the efficiency in terms of impact health.

## Figures and Tables

**Figure 1 ijerph-20-02649-f001:**
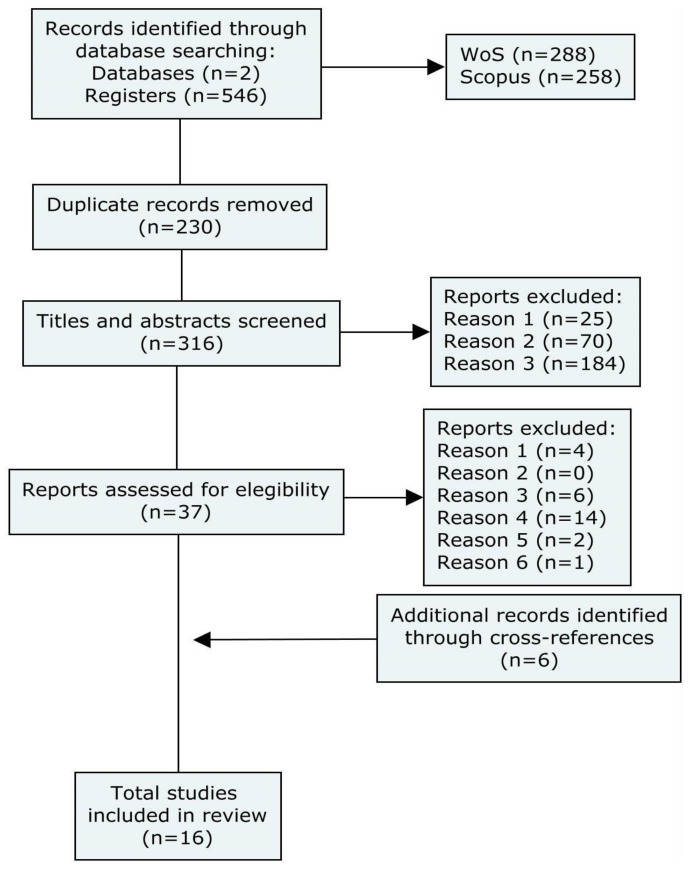
Scoping review search flowchart.

**Table 1 ijerph-20-02649-t001:** Descriptive data of the study variables in the articles.

Variable	Category	Percentage
Impact on health	Yes	81.25
	No	18.75
Transfer	Yes	43.75
	No	56.25
Post Evaluation	Yes	50
	No	50
Measure applied	A	37.5
	B	18.75
	C	31.25
	D	12.5
Push or pull	Push	75 ^1^
	Pull	37.5 ^1^

^1^ Some papers assessed both push and pull measures.

## Data Availability

Not applicable.

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
