# Peer review of "Effectiveness of Mobility and Urban Sustainability Measures in Improving Citizen Health: A Scoping Review"

_ijerph, 2023, doi:10.3390/ijerph20032649_

Round 1

Reviewer 1 Report

The article," Effectiveness of mobility and urban sustainability measures in 2 improving citizen health: a scoping review"

presented their review work as well defined manner and while,i gothrough the paper, it attracts me a lot for thier work, flow and concepts

-       In my concern, there is no novelty of the proposed work /the authors failed to justify the novelty of the work .
-       The topic is not so relevant on the field.

-       The authors are suggested to compare the previous method with their proposed work and able to justify the uniqueness of the proposed work

-       Limitations and the drawbacks of the existing work has been compared with the proposed work and show their innovation.

-       Conclusion parts are quite ok, but needs to address more on the clarity on the proposed work

-       Add to the latest references with the concise one.
-       Figures are ok. But proposed work needs more attention on the figuring

Author Response

Dear reviewer, 

First of all, the authors would like to thank you for the effort and work dedicated to the review of the manuscript, as well as for your comments, recommendations and requests for changes, which undoubtedly improve and enhance the quality of our work. To this end, we respond to each of them below, together with the changes made to the manuscript. 

Firstly, to indicate the innovation and justification for the novelty of the study, the same has been developed throughout the study, with the changes incorporated through the tool, given that in recent years, the literature does not offer any review of the set of measures, together with their respective analyses, applied in the different countries and communities, despite the importance of knowing which measures are being applied and the benefits they bring. We believe that this may shed light on a field still to be discovered. In the same way, an abstract has been developed with greater fluency, in order to improve the objective that it is intended to achieve. 

Regarding the comparison of the previous method, despite the fact that the authors present several doubts as to the method referred to, we believe that it is suggested to develop a better explanation of why it was decided to opt for the method of the panoramic review. To this end, a reference to the suitability of the scoping review in the field of health has been included in the article. For those complex multidimensional phenomena whose understanding requires multivariate analysis, conceptual and methodological development, and complementarity of sources, as well as the need to know what type and amount of evidence exists in a specific area, and whether it is feasible or necessary to conduct a systematic review in that area, the method of the scoping review arises, which provides an answer to these questions. 

Finally, with regard to the conclusions, we have sought to develop and clarify them further, with a more exhaustive and clearer wording, thereby seeking to show the final conclusions reached by the authors after the study. 

To conclude, we would like to thank you once again for your work, and we hope that the answers and changes given are up to the quality of the review and requests. We remain at your disposal should you wish to make any additional changes or clarifications to those already made. 

Yours sincerely, 

The authors.

Reviewer 2 Report

Your paper on "Effectiveness of mobility and urban sustainability measures in improving citizen health: a scoping review" is a good start toward a strong scope paper and it will be gaining strength if you go for less description and more synthesis and analysis of findings. Here are more concrete considerations where you could add some deeper reflections to your scope paper:

1.Methodologically you are following a good process to select and reduce your focus to 16 papers. However, after you have made this selection, it will be important to be more transparent about how, very specifically, do you understand "public health effectiveness" and what potential the relationship between urban/mobility, sustainability and impacts on health effectiveness is. What the literature says about the type of particular health impacts to be expected and whether "health" is a factor that motivates urban/mobility decision-makers to implement particular policies. Without a clear understanding of what aspects of health are affected by mobility/urban living, then we do not know what type of studies apply or not.  For instance, traffic safety and reduction of accidents are not included 

2. Why did you focus only on measures (A)promotion of bike use, (B) walkability (C) reduction optimization of car use and (D) mixed measures?  I did not find a good justification for why these classification? 

3. Section 3.2 is not a synthesis but mostly reads as a summary of the papers. You can do more to group the papers in accordance to your four categories and explain what do they contribute- are some measures more effective? A real analysis is here missing.

4. Your conclusions should not be a collection of bullet points. The paper presentation already indicates that more reflection is possible. Also, you need to answer very concretely your research question in your conclusion, if you can not provide a satisfactory answer then re-state why is so, and explain what the results allow to say.

Author Response

Dear reviewer, 

First of all, the authors would like to thank you for the effort and work dedicated to the review of the manuscript, as well as for your comments, recommendations and requests for changes, which undoubtedly improve and enhance the quality of our work. To this end, we respond to each of them below, together with the changes made to the manuscript. 

On the one hand, on the review carried out on the basis of the methodology and the development of health effectiveness and sustainability, with the aim of to make this part of the work clearer, the following improvements have been implemented: The research question has been rephrased using more specific terms that are included in international conventions ("public health impact") so that other researchers who are not familiar with these concepts can refer to these commonly accepted definitions if they have any doubts. Changes have been made in section 2 to clarify the central theme that guides authors in identifying relevant papers for the review.

With regard to the selection of the measures analysed in the study, it should be noted that this has been carried out because they are the set of measures identified in the studies selected in the overview review. These measures were grouped in their entirety in the four measures developed, A, B, C and D. To improve and clarify the justification for this grouping, a justification for their use has been included in the article. 

On the other hand, with regard to the recommendations in point 3.2, the authors would like to emphasise that this section is one of the most important parts of the scoping review methodology, in which an exhaustive synthesis of the most important results of the selected articles should be incorporated, which can be checked in turn in the tables indicated as appendices. Also indicate that the division made for the synthesis is due to the fact that some of the studies incorporate different measures, or a mixture of several of them. Due to this inherent limitation of the studies analysed, it is for what, in turn, a differentiation is carried out between the type of push and pull measures. We believe that carrying out this type of analysis can be a very effective way and a very interesting future line of research and/or advancement of the present one.

Finally, with regard to the conclusions, we have sought to develop and clarify them further, with a more exhaustive and clearer wording, thereby seeking to show the final conclusions reached by the authors after the study. All these changes can be seen in the new manuscript, incorporated through the platform.

To conclude, we would like to thank you once again for your work, and we hope that the answers and changes given are up to the quality of the review and requests. We remain at your disposal should you wish to make any additional changes or clarifications to those already made. 

Yours sincerely, 

The authors.

Reviewer 3 Report

Dear authors,

Thank you very much for your submission. Kindly find below some comments.

The title should be more specific. Is not clear where the research conducted - from which places you gathered data.

chapter 2.1 is only one sentence. No reason for that. You can merge it with part 2.2

part 2.2. must be more analytical and explain all the processes. Explain much more further the use of python, why you chose this process and how the results came out. 

Regarding the ''2 experts''. This process must be explained more scientifically.  Did they create specific reports, in which way you used their ''opinion''. 

There is a luck of understanding in the way you have worked and if scientific methods were used. 

Chapter 3 Results:

A major part of the results are more likely a literature review without focusing in your research results. Several changed need to be applied

Appendix part is extremely big. Consider to transfer some important info in the main part. 

Author Response

Dear reviewer, 

First of all, the authors would like to thank you for the effort and work dedicated to the review of the manuscript, as well as for your comments, recommendations and requests for changes, which undoubtedly improve and enhance the quality of our work. To this end, we respond to each of them below, together with the changes made to the manuscript. 

Firstly, with regard to the title, we cannot indicate a specific research location, as it is a panoramic review of the literature, analysing studies from around the globe. In it, we find studies from Asia, Europe, America, etc. This is why the title reflects the inclusion of scoping review. Regarding point 2.1, we should keep it, since it is one of the 6 parts of the scoping review methodology used in the study, as indicated in the methodology, " The scoping review was carried out following the methodology proposed by Arksey and O'Malley [14], which is used as a reference in the field ". However, we have reworded point 2.1 within the manuscript to make the inclusion of this section clearer.

With regard to the description of the Python process, it should be noted that, as the article reflects, it has been used to download and group the studies analysed in the literature review. We therefore consider that the description provided in the article is sufficient to explain this process, “Python language was used to process the files with the bibliographic records. Duplicates were removed to obtain a preliminary selection of studies”. The rest of the methodology is developed in the rest of the manuscript.

We have proceeded to develop and clarify a little more the methodological process of the overview review, with respect to the use of experts and reviewers of the studies, in order to clarify this section and that the understanding of the same is understood in a better way. 

With regard to the results, the authors would like to point out that this was precisely the aim of the study, given that it consists of a review of the literature, as indicated, and therefore the results section aims to show, in a synthesised and ordered manner and on the basis of the variables analysed, the results of the research and studies collected in this overview review of the literature. This is also why the appendices are so extensive, given the inherent nature of this type of work. However, we would like to point out that all the information contained in the appendices is written, synthesised and included throughout the manuscript.

To conclude, we would like to thank you once again for your work, and we hope that the answers and changes given are up to the quality of the review and requests. We remain at your disposal should you wish to make any additional changes or clarifications to those already made. 

Yours sincerely, 

The authors.

Reviewer 4 Report

Comments:

The authors established a review of the effectiveness of mobility and urban sustainability measures in improving citizen health in the manuscript. The idea is interesting. However, the significant observations of the reviewer are listed below, which must be addressed before publication.  

1. The Abstract section should be concise and informative and have flow. ‎ Moreover, the review scope and ‎rationale ought ‎to ‎be ‎included as well. ‎

2. Please, provide concise descriptions of the dataset, developed models, and the experimental procedure in aspects of brief review effectiveness of mobility and urban sustainability measures in improving citizen health. What sort of amendments, directions, improvements, etc.? What is the criterion for improving citizen health? Please provide a brief review of it.

3. This manuscript lacks the connection between the reviewed models, techniques, solutions, etc., to address the issue. ‎Please add more explanations, create a connection between these techniques, and give solid reasoning why this review is conducted and what directions are unfolding for researchers.

4. Compared this review with other literature reviews, what are the difficulties and challenges for the developed edge technologies in this area?

5. What are the limitations of this review?

6. In the conclusion part, the author should point out the next research direction and the parts that need to be further unfolded and make critical remarks.

Author Response

Dear reviewer, 

First of all, the authors would like to thank you for the effort and work dedicated to the review of the manuscript, as well as for your comments, recommendations and requests for changes, which undoubtedly improve and enhance the quality of our work. To this end, we respond to each of them below, together with the changes made to the manuscript. 

First, indicate that the wording of the abstract has been modified, seeking greater fluidity and clarity in terms of the objective, results and conclusions of the study and a more connected and striking writing and reading.

On the one hand, regarding the review carried out on the basis of the methodology and experimental design, we would like to note that descriptions of the datasets, the models developed and the experimental set-ups of the selected papers can be found in Appendix C, among other relevant information such as the measurements analyzed in each case. An improvement in public health is defined as an improvement in any public health indicator, including both direct and indirect measures.  To make this part of the work clearer, the following improvements have been implemented:

The research question has been rephrased using more specific terms that are included in international conventions ("public health impact") so that other researchers who are not familiar with these concepts can refer to these commonly accepted definitions if they have any doubts. Changes have been made in section 2 to clarify the central theme that guides authors in identifying relevant papers for the review.

Regarding the techniques used, in order to develop a greater justification for the use of the panoramic review in this study, a reference to the suitability of the scoping review in the field of health has been included in the article. For those complex multidimensional phenomena whose understanding requires multivariate analysis, conceptual and methodological development, and complementarity of sources, as well as the need to know what type and amount of evidence exists in a specific area, and whether it is feasible or necessary to conduct a systematic review in that area, the method of the scoping review arises, which provides an answer to these questions.

Finally, with respect not only to the conclusions, but also to the limitations and barriers found during the study, as well as the future steps or lines of investigation that the present investigation may bring about, a modification of the conclusions has been made. We have sought to develop and clarify them further, with a more exhaustive and clearer wording, thereby seeking to show the final conclusions reached by the authors after the study. All these changes can be seen in the new manuscript, incorporated through the platform.

To conclude, we would like to thank you once again for your work, and we hope that the answers and changes given are up to the quality of the review and requests. We remain at your disposal should you wish to make any additional changes or clarifications to those already made. 

Yours sincerely, 

The authors.

Round 2

Reviewer 3 Report

   Dear Authors, 
Thank you very much for your comments and the updated version, well received and noted.

You can finally proceed with a last check regarding minor spelling and syntax errors. 

Author Response

Dear reviewer, thank you again for your work and review process, very valuable for us. We proceed to review the manuscript thoroughly, together with the specific translation, which will be reviewed in turn by an external native processor. 
Again, thank you for your comments and added value.
Authors.